# DNA Methylation Influences miRNA Expression in Gonadotroph Pituitary Tumors

**DOI:** 10.3390/life10050059

**Published:** 2020-05-13

**Authors:** Joanna Boresowicz, Paulina Kober, Natalia Rusetska, Maria Maksymowicz, Agnieszka Paziewska, Michalina Dąbrowska, Natalia Zeber-Lubecka, Jacek Kunicki, Wiesław Bonicki, Jerzy Ostrowski, Janusz A. Siedlecki, Mateusz Bujko

**Affiliations:** 1Department of Molecular and Translational Oncology, Maria Sklodowska-Curie National Research Institute of Oncology, 02-781 Warsaw, Poland; joanna.boresowicz@gmail.com (J.B.); paulina.kober@pib-nio.pl (P.K.); natarusetska@gmail.com (N.R.); jas@coi.waw.pl (J.A.S.); 2Department of Pathology and Laboratory Diagnostics, Maria Sklodowska-Curie National Research Institute of Oncology, 02-781 Warsaw, Poland; Maria.Maksymowicz@pib-nio.pl; 3Department of Genetics, Maria Sklodowska-Curie National Research Institute of Oncology, 02-781 Warsaw, Poland; agapaziewska@poczta.onet.pl (A.P.); Michalina.Dabrowska@pib-nio.pl (M.D.); jostrow@warman.com.pl (J.O.); 4Department of Gastroenterology, Hepatology and Clinical Oncology, Medical Center for Postgraduate Education, 01-813 Warsaw, Poland; natalia.zeber@o2.pl; 5Department of Neurosurgery, Maria Sklodowska-Curie National Research Institute of Oncology, 02-781 Warsaw, Poland; jacek.kunicki@pib-nio.pl (J.K.); neurochirurgia@coi.waw.pl (W.B.)

**Keywords:** epigenetics, DNA methylation, microRNA, gene expression, PitNET, gonadotropinoma

## Abstract

microRNAs are involved in pathogenesis of cancer. DNA methylation plays a role in transcription of miRNA-encoding genes and may contribute to changed miRNA expression in tumors. This issue was not investigated in pituitary neuroendocrine tumors (PitNETs) previously. DNA methylation patterns, assessed with HumanMethylation450K arrays in 34 PitNETs and five normal pituitaries, were used to determine differentially methylated CpGs located at miRNA genes. It showed aberrant methylation in regions encoding for 131 miRNAs. DNA methylation data and matched miRNA expression profiles, determined with next-generation sequencing (NGS) of small RNAs, were correlated in 15 PitNETs. This showed relationship between methylation and expression levels for 12 miRNAs. DNA methylation and expression levels of three of them (*MIR145*, *MIR21*, and *MIR184*) were determined in the independent group of 80 tumors with pyrosequencing and qRT-PCR and results confirmed both aberrant methylation in PitNETs and correlation between methylation and expression. Additionally, in silico target prediction was combined with analysis of established miRNA profiles and matched mRNA expression pattern, assessed with amplicon-based NGS to indicate putative target genes of epigenetically deregulated miRNAs. This study reveals aberrant DNA methylation in miRNA-encoding genes in gonadotroph PitNETs. Methylation changes affect expression level of miRNAs that regulate putative target genes with tumorigenesis-relevant functions.

## 1. Introduction

Gonadotroph pituitary neuroendocrine tumors are among the most frequently diagnosed tumors of the pituitary gland. The vast majority of these tumors are clinically nonfunctioning. In contrast to most pituitary tumors developed from pituitary cells, other than gonadotropes they do not cause any characteristic endocrinological symptoms [1]. The biology of gonadotroph PitNETs (pituitary neuroendocrine tumors) is a matter of current research and epigenetic abnormalities seem to be one of important features of tumorigenesis of pituitary cells [2].

MicroRNAs are small RNA (ribonucleic acid) particles that regulate gene expression at post-transcriptional level by interacting with 3′ UTR (untranslated region) sequence of target mRNA which is determined by complete or incomplete sequence complementarity. In general, binding mRNA (messenger RNA) untranslated region by miRNA (microRNA) incorporated in RISC complex results in its degradation or blocking the translation. Thus, miRNA particles are generally considered to be negative regulators of expression [3].

Alterations of miRNA expression are one of the hallmarks of cancer and changes in miRNA expression affect transcriptomic profile of tumor cells. Tumor-related miRNAs may be classified as onco-miRNAs when their expression enhances tumor growth or suppressive miRNA when they negatively regulate oncogenic proteins. A growing body of evidence supports the role of miRNA in pathogenesis of pituitary tumors [4].

MiRNAs are encoded in human genome and similarly to protein-coding genes are transcribed mainly by RNA polymerase II. Primary miRNA transcript i.e., pri-miRNA undergoes further processing including cleavage to pre-miRNA form by Drosha/DGCR8 complex and subsequent export to the cytoplasm where it is cleaved by Dicer to generate the mature miRNA. Similarly to protein-coding mRNA, the transcription of pri-miRNA is epigenetically regulated and the role of DNA (deoxyribonucleic acid) methylation in miRNA expression is well documented [3]. Cancer-related aberrant methylation is commonly observed in genes encoding for miRNA which may contribute to impairment of miRNA expression and level of its target mRNA. A number of miRNAs that are epigenetically dysregulated in human cancers have been identified [5]. It is known that epigenetics plays a role in pathogenesis of pituitary tumors, however, the relationship between DNA methylation and miRNA expression in pituitary tumors has not been investigated.

In this study we identified CpGs located in miRNA genes that have differential methylation levels in gonadotroph PitNETs and normal pituitary and compared profiles of differential methylation and expression of miRNA in gonadotroph tumors. The identified miRNAs with expression level related to DNA methylation were further investigated for their potential role in tumor development.

## 2. Results

### 2.1. DNA Methylation Changes in miRNA Genes in Gonadotroph PitNETs

DNA methylation in miRNA-encoding genes was determined using the results of previous methylation profiling of 34 PitNETs and five samples of normal pituitary with HumanMethylation450K arrays (Illumina) [6]. Data analysis included 407,939 HM450K (HumanMethylation450 BeadChip Illumina arrays) probes after normalization procedure and excluding the probes at polymorphic positions and sex chromosomes as well as probes with lacking signal. This set of probes included 2844 probes covering CpGs located at miRNA encoding genes (in both regulatory and encoding sequences), according to Illumina annotation.

Differentially methylated probes, i.e., CpGs with significantly different methylation level were identified by comparison of gonadotroph PitNETs and normal pituitaries data. Using criterion of delta beta value >0.15 or <−0.15 that corresponds to at least 15% difference in DNA methylation level we found 228 CpGs annotated to miRNA-encoding genes methylated at different level in tumors and normal tissue. These DMPs (differentially methylated probes) are located in or near 131 miRNA-encoding genes. A total of 143 DMPs (located at 88 miRNA genes) were hypermethylated while 84 DMPs (47 miRNA genes) were hypomethylated in pituitary tumors, as shown in Figure 1A. Most of DMPs (190 CpGs) are those located at 5′ upstream region while 38 CpGs are located in miRNA gene bodies (Figure 1B). When DMPs were classified according to CpG content 22 of them were found to be located in CpG islands, 33 in CpG island shelves/shores, and 44 were located outside islands (so-called open see CpGs) (Figure 1B). Differentially methylated CpGs are listed in detail in Appendix A.

### 2.2. The Relationship between Impaired Methylation and Expression of miRNA Genes

Fifteen gonadotroph PitNET samples with previously determined DNA methylation profile were subjected to miRNA expression analysis with next-generation sequencing of small RNA fractions using semiconductor sequencing technology. Sequencing of small RNA libraries generated an average of 2,497,367 reads per sample, which were mapped to the human genome (hg19) and used for quantification of expression levels of known miRNAs. Sequencing reads were annotated to 1528 miRNAs and after filtering out those with low expression, the measurements of 405 miRNAs were used in the analyses of the relationship between DNA methylation and miRNA expression.

The results of methylation profiling of 15 gonadotropinoma samples with HM450K arrays (i.e., normalized B-values for miRNA-related DMPs) and matched normalized read counts for particular miRNAs (determined with small RNA sequencing) were used to determine the miRNA-related DMPs for which DNA methylation level is correlated with the expression of the particular miRNAs. Spearman correlation analysis was used.

Significant correlation was observed for 20 miRNA-related DMPs that are located at 12 miRNA-encoding genes. Nineteen expression-correlated DMPs were located in regions upstream transcription start site including 13 CpGs located in regions 1–200 bp upstream TSS (transcription start site) and six CpGs in regions located between 201 and 1500 bp upstream TSS. Only one transcription-correlated DMP was located in miRNA-encoding sequence. A negative correlation between expression and DNA methylation level was observed for seven miRNA encoding genes including hsa-miR-589-3p, hsa-miR-145 (both hsa-miR-145-5p and hsa-miR-145-3p), hsa-miR-150-5p, hsa-miR-21-5p, hsa-miR-184, hsa-miR-23B-3p, and hsa-miR-590-3p. On the other hand, a positive DNA methylation/expression correlation was found for the following five miRNAs: hsa-miR-134-5p, hsa-miR-141-5p, hsa-miR-487A-3p, hsa-miR-489-3p, and hsa-miR-758-3p. Details are presented in Appendix A.

### 2.3. Validating the Role of Selected Aberrantly Methylated miRNAs in a Large Patient Cohort

To verify the results of correlation analysis based on methylation arrays and sequencing data we determined DNA methylation level at selected identified CpGs with pyrosequencing assays and expression of corresponding miRNAs with qRT-PCR (real-time quantitative reverse transcription PCR). This validation step was performed on gonadotroph PitNET samples from the independent cohort of 80 patients. DNA methylation was determined with bisulfite pyrosequencing of promoter regions of three selected miRNA-encoding genes, *MIR145*, *MIR21*, and *MIR184*. *MIR145* and *MIR21* were chosen for validation procedure since both have a documented role in pituitary tumors. *MIR184* was found as overexpressed in PitNETs. Results confirmed promoter hypermethylation of *MIR145* and *MIR21* as well as hypomethylation of promoter region of *MIR184* in pituitary tumors in comparison to normal tissue, as shown in Figure 2.

To verify the identified methylation/expression relationship we ran a correlation analysis using pyrosequencing measurement of methylation level at a particular CpG site and level of corresponding miRNA expression in the independent validation group of samples.

Expression levels were determined for three selected miRNAs: hsa-miR-145-5p, hsa-miR-21-5p, and hsa-miR-184 with qRT-PCR in 80 PitNET samples. To determine a suitable reference gene for normalization of qRT-PCR data four small RNAs were tested: 5s RNA, U6 RNA, *SNORD44*, and *SNORA66* that are commonly used as reference. The stability of these RNAs expression was tested in PitNET samples, and their usefulness as reference genes was assessed with RefFinder [7]. The results are shown in Appendix A. The expression level of *SNORA66* was below qPCR (quantitative PCR) detection level in the large proportion of samples and was not included in the analysis of expression level stability. Based on the examination of candidate references the geometric mean of Ct values from two references *SNORD44* and 5s RNA was used as reference in calculation of qPCR results.

Analysis of correlation between DNA methylation measured with bisulfite pyrosequencing and miRNA expression levels determined with qRT-PCR was performed for four CpG sites located at three genes encoding for hsa-miR-184, hsa-miR-145-5p, hsa-miR-21-5p. As a result, we observed a significant methylation/expression correlation for all analyzed miRNAs confirming the results from the discovery group. Results are presented in Figure 1B and Table 1.

### 2.4. Predicting the mRNA Targets of Epigenetically Deregulated miRNAs

Since the role of miRNA is the regulation of mRNA it is believed that functional consequences of changed miRNA level would be reflected by the change in the expression of its target mRNA. We looked for mRNAs that potentially interact with the identified miRNAs, whose expression appeared to be related to aberrant methylation in gonadotroph PitNETs. For this purpose, in silico mRNA target prediction was combined with correlation analysis of miRNA and mRNA quantification results as described in detail in Section 4. The matched data from small RNA sequencing and previous mRNA sequencing of gonadotroph tumor samples with Ion AmpliSeq™ Transcriptome Human Gene Expression Kit (Thermo Fisher Scientific) was used. The analysis was performed for 19 miRNAs that have been identified as abnormally methylated in gonadotroph tumors, with methylation-related expression level. After filtering out genes with low expression, the measurements of 12,778 mapped human mRNAs (approx. 61% of the Ion AmpliSeq Transcriptome) were available for correlation analysis.

At first, we determined the predicted miRNA–mRNA interactions with MirDIP algorithm [8] with the use of “Very High” MirDIP prediction score as a criterion. Subsequently, the correlation between the expression levels of potentially interacting miRNA and mRNA was determined with the use of data from miRNAome and transcriptome profiling of gonadotroph PitNET samples. The analyzed miRNAs have multiple predicted targets and when the adjustment for multiple testing was applied in correlation analysis the results were below significance threshold level. However, since Spearman R coefficient provides also important information on the quality of results and the correlation is only the part of the identification of potential miRNA–mRNA interactions, we used a non-adjusted *p*-value to present the results.

The procedure allowed for identification of 56 genes potentially regulated by miRNAs that may be considered epigenetically misregulated in tumors. The identified putative interacting miRNA–mRNA pairs are listed in Appendix A.

Using two gene ontology catalogs—KEGG pathways and GO biological processes—we examined whether any particular pathways are enriched for the identified genes. Unfortunately, we did not find any significant enrichment with adjusted *p*-value < 0.05.

To further interpret the results of our correlation-based miRNA-mRNA interactome analysis we analyzed the list of 56 potential target mRNAs in search of known cancer-related genes. For this purpose, we examined whether the identified genes are within the catalog of cancer genes. The Network of Cancer Genes 6.0 (NCG) is a literature-based recently updated database of known and predicted cancer driver genes [9]. Six potentially miRNA-interacting mRNAs have been found in NCG atlas. They are predicted targets of five miRNAs, that can be treated as epigenetically deregulated putative onco-miRNAs, as shown in Table 2.

Additionally, as a result of our own literature review of the function of predicted targets we classified the identified putative interactions between selected miRNA-mRNAs as potentially relevant in terms of tumor pathogenesis or pituitary functioning. This includes pairs of hsa-miR-23b-3p and *NACC1*, a cancer-related transcription regulator [10], hsa-miR-23b-3p and *PIK3R3*, as well as hsa-miR-134-5p and *INA* gene that encodes neuronal intermediate filament involved in pituitary tumorigenesis [11].

Additionally, by the search of miRTarbase that contains evidence of experimentally confirmed miRNA–mRNA interactions we determined which of our predicted miRNA-mRNA pairs have been confirmed in previous experimental research. The experimental evidence of eight miRNA–mRNA interactions is available in miRTarbase including four interactions with mRNAs of tumor-related genes. The list of these putative onco-miRNAs with predicted and expression-correlated target genes, is presented in Table 2, while the complete list of 10 differentially methylated miRNAs, for which the predicted targets were identified is presented in Appendix A.

## 3. Discussion

The role of epigenetic dysregulation of miRNAs was observed in many human cancer types but it was not investigated in tumors of the pituitary gland previously. Since miRNAs are transcribed in a similar way as protein-coding genes, epigenetic mechanisms, including DNA methylation, play a role in the regulation of the expression of miRNA-encoding genes [5,12]. This study aimed to determine what miRNA-encoding genes have different DNA methylation levels in pituitary tumors and normal pituitary samples, to investigate whether differential methylation affects miRNA expression and what could be the probable consequences for mRNA expression. The study was focused on particular subtype of the pituitary tumors—gonadotroph nonfunctioning PitNETs.

DNA methylation microarrays were used for determining methylation level at CpG sites located at miRNA-encoding sequence and approximately 8% of these CpGs located at 131 miRNA genes were found differentially methylated in tumors compared to normal pituitary sections. Subsequent correlation-based analysis revealed that expression of 12 miRNAs is related to CpG methylation. Negative correlation, where higher methylation is related to decreased miRNA expression, was observed for seven miRNAs. For five of them, including hsa-miR-145 [13], hsa-miR-150 [14], hsa-miR-23b [15], hsa-miR-589 [16], and hsa-miR-21 [17] the expression changes were found as related to aberrant DNA methylation in different human tumors. In turn, positive correlation, where increase in methylation level relates to enhanced miRNA expression, was found for five miRNAs including hsa-mir-141 where epigenetic deregulation was found in prostate cancer [18]. DNA methylation was generally linked to silencing of gene transcription, however, it has been recently observed that high methylation at genomic sequences flanking pre-mRNA can augment the expression of miRNA [12] which can explain the observation of positive correlation between methylation and expression of miRNAs in pituitary tumors.

Most of these identified miRNAs show aberrant expression levels in human cancers according to OncomiR database [19] and most of them have well described tumor driving or inhibiting functions in tumorigenesis. Hsa-miR-145 is a known tumor suppressor, downregulated in multiple types of cancer which is involved mainly in the regulation of invasion and migration [13]. Similarly hsa-miR-489 is considered as tumors suppressive miRNA [20,21,22]. In turn, hsa-mir-21 was manly reported as oncogenic miRNA which is manly overexpressed in many cancers where it positively regulates proliferation and invasion and inhibits apoptosis [23]. A characteristic feature of miRNAs is that they play opposite roles in developing cancers of different tissues of origin or different histology. Commonly, the same miRNA act as oncogene in one cancer type and tumor suppressor in another one. Sometimes the contradictive functions are reported for the same cancer type. Hsa-miR-134 belongs to such “ambiguous” miRNAs. It was found as downregulated in some cancers and upregulated in another ones with even contradictory results in studies on the same type of lung cancer [23]. Similarly opposite functions in tumorigenesis were reported with regard to the other miRNA that we found as epigenetically misregulated, including hsa-miR-141 [24], hsa-miR-23b [25], has-mir-150 [26], hsa-miR-184 [27,28], or hsa-miR-590 [29,30].

Over half of miRNA with DNA methylation related expression that we report have been previously described as differentially expressed in pituitary tumors including hsa-miR-23b [31], hsa-miR-145 [32], hsa-miR-150 [33], hsa-miR-141 [33,34], hsa-miR-21 [35], hsa-miR-134 [36], hsa-miR-184 [37,38].

DNA methylation and expression levels of three selected miRNA-encoding genes *MIR145*, *MIR21*, and *MIR184* were measured with distinct analytical methods, bisulfite pyrosequencing, and qRT-PCR, respectively, in additional tumor samples from an independent group of patients. The results were concordant with those obtained with the microarray and sequencing approach. Similar differences in DNA methylation between normal and tumor samples were observed and similar correlation of the five selected CpG sites and the expression levels of the associated miRNAs was found, providing validation for high-throughput correlation discovery analysis.

miRNAs act as regulators of genes expression, therefore, identification of the particular mRNAs which are the targets of miRNA particles is crucial for the interpretation of miRNA function. It is also required for the evaluation of possible functional consequences of impaired expression of particular miRNAs in tumors. In the common investigational approach potential target miRNAs are identified with in silico methods of target prediction and, subsequently, the interactions are verified in a cell-line model. This type of research has limited application when investigating pituitary tumors since very few cell-line models of pituitary adenomas are available and these cell lines are derived from mice and rodents. The only human-derived pituitary tumor cell line HP75 was used for in vitro testing and recently this cell line has been withdrawn from ATCC collection where it was previously available. In our study we made an attempt to identify targets for the identified epigenetically deregulated miRNAs by combining target prediction and analysis of correlation between expression levels of the particular miRNA and probable target mRNA. This approach has been utilized previously predicting mRNA targets for tumor-related miRNAs [39]. Although this procedure does not substitute in vitro functional testing for miRNA–mRNA interaction, it allows for determination of probable target mRNAs in a high-throughput manner. As a result of analysis, we obtained a list of putative target genes of epigenetically deregulated miRNAs. For determining the role of particular targets, we searched the database of cancer-driving genes The Network of Cancer Genes 6.0 and the available literature. This allowed to determine what target genes potentially have a role in tumorigenesis and, accordingly, which of epigenetically deregulated miRNAs could be classified as putative onco-miRNAs. In general, these results should be treated with caution, however, part of the correlation-based target predictions seem to be reliable in the context of previous research.

The clearest results are for hsa-miR-145-5p which appears to be a tumor suppressive miRNA hypermethylated in gonadotroph PitNETs. In gonadotroph PitNETs its expression is correlated with two potential oncogenes, *MAP2K4* and *CDH2*, which are highly probable targets of this miRNA. These targets are predicted with a very high MirDIP score and the evidence for the direct interaction of miR-145-5p and mRNAs of these two genes has been published [40,41]. Hsa-miR-145 has a well-documented suppressive role in cancer development and its epigenetic downregulation was previously described in human prostate, breast, ovarian, and lung cancers [13]. Downregulation of hsa-miR-145-5p in nonfunctioning PitNETs was previously found in comparison of tumor and normal pituitary samples and its suppressive effect on proliferation and invasion was shown in cell culture and an xenograft model [32].

Additionally, the results showing hypermethylation at *MIR23B* in nonfunctioning PitNETs and its inverse correlation of methylation and expression levels are concordant with previously published data showing downregulation of this miRNA in gonadotropinomas [31]. The authors showed the suppressive role of hsa-miR-23b that acts as inhibitor of proliferation by interaction with *HMGA2* transcript [31]. Our correlation-based target predictions indicate that regulatory unit of PI3K i.e., *PIK3R3* and cancer-driving transcription factor *NACC1* are probable targets of hsa-miR-23b. Both putative target genes have an effect on cell proliferation [42,43].

Our data also suggest that epigenetically deregulated hsa-miR-134-5p may play a particular role in neoplastic transformation of pituitary cells. *MIR134* is hypomethylated in PitNETs with positive methylation/expression correlation and it is a putative regulator of *INA*. This gene encodes for internexin neuronal intermediate filament protein alpha. According to GTEx database [44] expression of this gene is specific to pituitary and brain and the role of the encoded protein in development of PitNET was previously indicated [11].

In summary, both aberrant epigenetic regulation and changes of miRNA expression play a significant role in pathogenesis of PitNETs by deregulating transcriptomic profile that affects important signaling pathways [2,4]. Our results show an interplay between these two regulatory layers in gonadotroph PitNETs where tumor-related DNA methylation contributes to miRNA expression.

## 4. Materials and Methods

### 4.1. Patients and Samples

Pituitary tumor tissue samples from 95 patients were collected during transsphenoidal surgery and immediately frozen in liquid nitrogen for storage at −80 °C. Part of each tumor was assessed histopathologically.

All the patients were treated by the same neurosurgeon according to the same surgical protocol. Samples were collected in Maria Sklodowska-Curie National Research Institute of Oncology in Warsaw between 2010 and 2016 and the diagnoses were based on WHO 2004 criteria, applicable at the time of tissue collecting [17]. Histopathological investigation included both immunohistochemistry and ultrastructural investigation. The study included gonadotrophic clinically non-functioning tumors that represent the most common histopathological subtype of PitNETs. Tumor samples were divided into two groups: discovery (15 samples) that was subjected for high-throughput profiling of DNA methylation, small RNA sequencing and transcriptome sequencing and validation group (80 samples) that was used for validation of the results obtained for selected miRNAs.

Agreement from Local Ethical Committee was obtained for the use of human tissue for the particular scientific purpose (decision no. 20/2019) and each patient provided informed consent for the use of tissue samples for scientific purposes. Patient profiles are presented in Table 3. Detailed patients’ clinical data are presented in Appendix A.

Five samples of normal human pituitary from autopsies were used for DNA methylation profiling. The samples underwent hematoxylin/eosin staining and histopathological evaluation to exclude the presence of incidental pituitary tumors. Postmortem interval for the samples was in the range of 25–46 h (median 35.5). According to previously reported data this PMI (postmortem interval) does not affect DNA methylation assessment with bisulfite-based methods [45], but samples with this PMI are not suitable for reliable expression analysis profile [46,47], thus, normal pituitary samples were used for DNA methylation only.

DNA was isolated with Qiamp DNA mini (Qiagen, Venlo, Netherlands) while total RNA was isolated with MirVana miRNA Isolation Kit (Thermo Fisher Scientific, Waltham, MA, USA). DNA and RNA quality was assessed spectrophotometrically using NanoDrop 2000 (Thermo Fisher Scientific). Isolated RNA was stored at −80 °C until analyses.

### 4.2. Profiling Genome-Wide DNA Methylation with Microarray Technology

Previously obtained results of genome-wide DNA methylation analysis with HumanMethylation450 BeadChip (HM450K) (Illumina) microarrays in gonadotroph pituitary tumors and normal pituitary sections were used [18]. This dataset includes methylation profiles of 34 tumor and five normal pituitary samples and is deposited at Gene Expression Omnibus under accession number GSE115783. HM450K array covers 3439 CpG sites located at 727 miRNA genes.

Probes differentially methylated between pituitary tumors and samples of normal pituitary gland were identified using ChAMP data analysis pipeline [19] as described previously [18]. Delta β-value was used as a measure of DNA methylation difference between tumor and normal tissue and was calculated by subtracting mean β-value of normal samples from mean β-value of tumor samples.

Differentially methylated probes were defined as those with delta β-value >0.15 or <−0.15 and adjusted *p* > 0.05. Microarray probes were annotated to human genome using IlluminaHumanMethylation450k.db library. DMPs located in genomic regions encoding for miRNAs were extracted for further analysis.

### 4.3. Assessment of miRNA Expression with Next Generation Sequencing (NGS)

Sequencing of small RNA fraction was done with semiconductor sequencing technology. Fifteen gonadotroph pituitary tumor samples with previously determined DNA methylation profile were included. Total RNA (25 ng) of each sample was subjected to construction of a small RNA sequencing library with Ion Total RNA-Seq Kit v2 (Thermo Fisher Scientific) according to manufacturer’s protocol. Ion Xpress™ RNA-Seq Barcode Kit was used for ligation of RNA adapters that allows for multiplexed sequencing. Amount and size distribution of DNA in the sequencing library was determined using Bioanalyzer 2100 using a High Sensitivity DNA Kit (Agilent, Santa Clara, CA, USA). Ion Chef instrument with Ion 318™ Chip Kit v2 BC was used for library preparation. Ion Torrent PGM sequencer (Thermo Fisher Scientific) with 500 run flows was used for sequencing. Unmapped bam files were converted into fastq files with a bamToFastq script from bedtools. Mapping sequence read to the human genome (hg19), quantification of known miRNA (according to miRBase release 21) was performed using miRDeep2.14

The results of miRNA sequencing were used for the analysis of correlation between DNA methylation at CpGs located in miRNA-encoding genomic regions and expression levels of corresponding miRNAs. Spearman correlation of methylation beta values normalized using the Beta-Mixture Quantile (BMIQ) method and normalized read count values from sequencing small RNA libraries was calculated in R environment.

### 4.4. Evaluation of DNA Methylation Pattern at Selected Genomic Regions

Pyrosequencing of bisulfite-treated DNA was used to determine DNA methylation at particular miRNA-encoding genomic regions. Conversion of 1 µg of DNA was performed with EpiTect kit (Qiagen). Bisulfite-treated DNA was PCR (polymerase chain reaction) -amplified in 30 µL volume containing 2 mM MgCl_2_, 0.25 mM dNTPs, 0.2 μM of each primer, and 0.5 U of FastStart DNA Polymerase (Roche Applied Science, Mannheim, Germany) with following temperature cycles: 94 °C for 3 min; 40 cycles of 30 s at 94 °C. 30 s at 55 °C, and 30 s at 72 °C; and a final elongation for 7 min at 72 °C. Then, PCR amplicons were purified and analyzed using the PyroMark Q24 System (Qiagen) according to manufacturer’s protocol. PCR primers are presented in Appendix A.

### 4.5. Determining miRNA Expression Level with qRT-PCR

Measuring the expression levels of selected mature miRNAs was done with MiRCURY™ LNA™ miRNA PCR System (Qiagen). Total RNA (50 ng) was subjected to reverse transcription using miRCURY LNA™ Universal RT miRNA PCR and cDNA synthesis kit (Qiagen) and subsequently diluted ×25. A total of 5 µL of PCR reaction contained 2 µL of diluted cDNA, 1 × SYBR Green Master Mix (Qiagen) and MiRNA LNA PCR primerset (Qiagen) and was run in 384-well format using 7900HT PCR system (Applied Biosystems, Foster City, CA, USA). qRT-PCR assays are listed in Appendix A. All qPCR measurements were performed in technical triplicates.

### 4.6. miRNA Target-Prediction

A two-step procedure was applied for the identification of possible mRNA targets of miRNA that were found to be aberrantly DNA methylated in gonadotroph pituitary tumors. At first, potential target mRNA were identified with MirDIP algorithm that combines multiple sources for miRNA target predictions [8]. Then, the correlation between the expression levels of identified potentially interacting miRNAs and mRNAs was assessed.

For this purpose, data from small RNA sequencing of PitNET samples and matched data from whole transcriptome next generation sequencing for the same tumor samples were used. Expression of protein-coding genes was determined with Ion AmpliSeq™ Transcriptome Human Gene Expression Kit (Thermo Fisher Scientific) and semiconductor sequencing technology. The method allows for measurement of approximately 20,000 human genes in a single assay. The results of mRNA amplicon-based sequencing were generated for previous research and details of preparing human transcriptome sequencing library and sequencing procedure were published previously [48]. Data have been deposited in Gene Expression Omnibus database under accession number GSE136781. The raw unnormalized count matrix was generated from BAM files using the GenomicAlignments package [49] and imported to DESeq2 for data normalization and calculation of the estimates of dispersion [50]. Low-expression gene filtration was applied (genes with at least five sequencing reads in at least half of the samples were included). Correlation between normalized read counts for particular miRNAs from small RNA sequencing and normalized read counts for predicted target genes was calculated using the Spearman method in the R environment.

Additionally, the putative miRNA–mRNA interactions identified by combined target prediction and correlation analysis were tested for the evidence of experimentally validated interactions using miRTarbase [51]. In contrast to MirDIP which is the target prediction tool, miRTarbase provides the complementing information on experimental results for predicted miRNA–mRNA pairs.

The Network of Cancer Genes 6.0 (NCG) [9] that represent a catalog of 2372 genes with known or predicted cancer driver roles was used to identify the putative miRNA targets that play a role in tumor development.

### 4.7. Statistical Analysis

Quantitative continuous variables were analyzed by a two-sided Mann–Whitney U-test and Spearman Correlation. A significance threshold α = 0.05 was adopted. *p*-value adjustment with Benjamini and Hochberg method was applied in differential methylation HM450K-based analysis, correlation analysis of miRNA expression and CpG methylation, correlation analysis of miRNA, and predicted target mRNA expression and gen set enrichment analysis. R environment and GraphPad Prism 5.0 (GraphPad Software) were used for analysis and data visualization.

## Figures and Tables

**Figure 1 life-10-00059-f001:**
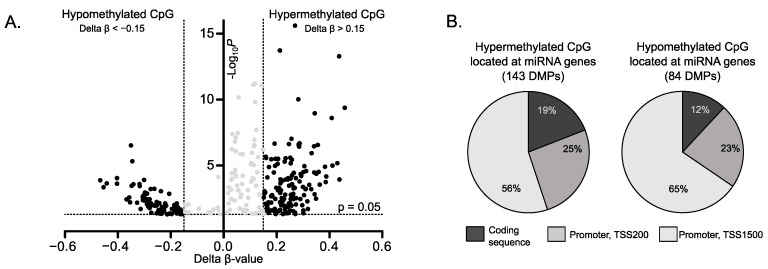
Distribution of CpGs located at miRNA-encoding genes that were differentially methylated in gonadotroph PitNETs and normal pituitaries. (**A**) Volcano plot of differentially methylated CpG sites. (**B**) Differences in the proportions of aberrantly methylated CpGs, stratified according to its position relatively to transcription start site (TSS).

**Figure 2 life-10-00059-f002:**
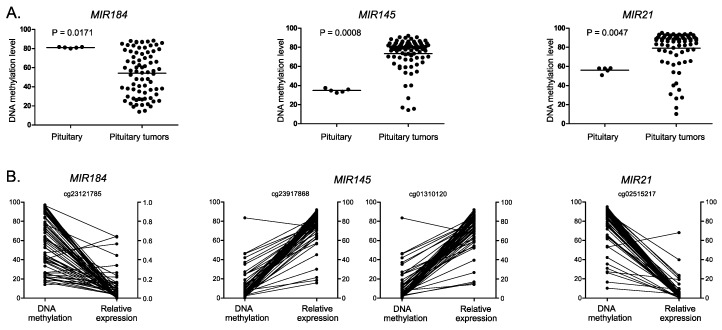
DNA methylation and relative expression of selected miRNA genes in the validation cohort. (**A**) Difference in DNA methylation at miRNA-encoding gene promoters in gonadotroph PitNET samples and normal pituitaries. Each dot represents the average methylation level of CpGs covered by PCR amplicon in a particular sample; mean values are horizontal lines. (**B**) Matched result of measurements of DNA methylation at particular CpG sites and relative miRNA expression levels. Each dot represents the CpG methylation/miRNA expression of individual PitNET sample.

**Table 1 life-10-00059-t001:** Comparison of the results of DNA methylation/gene expression correlation analysis from the study that used HM450K data and NGS-based transcriptomic profile (discovery group) and the validation study performed by pyrosequencing and qRT-PCR (validation group).

HM450K CpG Site	CpG Location	miRNA	Discovery Group Spearman R; adj. *p*-Value	Validation Group Spearman R; *p*-Value
cg23121785	TSS1500	hsa-miR-184	−0.577; adj. *p* = 0.0244	−0.524; *p* < 0.0001
cg01310120	TSS200	hsa-miR-145-5p	−0.608; adj. *p* = 0.0163	−0.312; *p* = 0.0061
cg23917868	TSS200	hsa-miR-145-5p	−0.672; adj. *p* = 0.0122	−0.448; *p* < 0.0001
cg02515217	TSS200	hsa-miR-21-5p	−0,796; adj. *p* = 0.0011	−0.336; *p* = 0.0030

**Table 2 life-10-00059-t002:** The results of mRNA target prediction and correlation analysis of miRNA and target mRNA expression levels in NFPA samples.

Putative Onco-miRNAs Based on Identification of Target mRNAs in NCG 6.0					
miRNA	DNA Methylation/Expression Results		Target Gene Prediction		
	Methylation status in tumor	DNA methylation/expression relationship	Putative target mRNA	Correlation (Spearman R, non-adjusted *p*-value)	Confidence of target prediction with MirDIP
hsa-miR-134-5p	*Hypomethylated*	*Positive*	*SCN9A*	−0.531, 0.0080	Very high
hsa-miR-145-5p	*Hypermethylated*	*Negative*	*MAP2K4*	−0.539, 0.0071	Very high
			*CDH2*	−0.515, 0.0100	Very high
hsa-miR-150-5p	*Hypermethylated*	*Negative*	*CACNA2D1*	−0.514, 0.0102	Very high
hsa-miR-21-5p	*Hypermethylated*	*Negative*	*RNF111*	−0.558, 0.0053	Very high
hsa-miR-23b-3p	*Hypermethylated*	*Negative*	*CCDC6*	−0.505, 0.0116	Very high
Putative onco-miRNAs based on literature-based interpretation of role of target mRNA					
**miRNA**	**DNA Methylation/Expression Results**		**Target Gene Prediction**		
	Methylation status in tumor	DNA methylation/expression relationship	Putative target mRNA	Correlation (Spearman R, non-adjusted *p*-value)	Confidence of target prediction with MirDIP
hsa-miR-23b-3p	*Hypermethylated*	*Negative*	*NACC1*	−0.510, 0.0107	Very high
			*PIK3R3*	−0.504, 0.0117	Very high
hsa-miR-134-5p	*Hypomethylated*	*Positive*	*INA*	−0.5103, 0.0107	Very high

**Table 3 life-10-00059-t003:** Clinical characteristics of PitNET patients.

Characteristic	Genome-Wide DNA Methylation/Whole Transcriptome Profiling	DNA Pyrosequencing/qRT-PCR
**PitNET patients (number of patients)**	15	80
Age (years)		
Range	36–73	34–80
Median	57	61
Gender (number of patients)		
Male	10	46
Female	5	34
Histopathology (number of patients)		
Gonadotroph PA	15	74
Null-cell/gonadotroph PA *	0	6
Clinical classification (number of patients)		
Invasive NFPA	5	49
Non-invasive NFPA	10	25
Unknown	-	6

* null-cell adenomas with clear ultrastructural gonadotroph features.

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
