# Peer review of "DNA Methylation Influences miRNA Expression in Gonadotroph Pituitary Tumors"

_life, 2020, doi:10.3390/life10050059_

Round 1
Reviewer 1 Report
Boresowicz and colleagues presented a very interesting research article aimed to determine the role of DNA methylation as an epigenetic mechanism responsible for the modulation of microRNAs expression levels in gonadotroph pituitary neuroendocrine tumors (PiTNETs). Through several experimental and computational approaches, the authors smartly elucidate the negative and positive correlations existing between DNA methylation (both promoter and intragenic methylation) and the expression levels of miRNAs detected in PiTNET samples. Overall, the manuscript is well written and the experimental design well-conceived, however, there are some minor/major revisions that the authors should address:
1) The title should be modified as follows: “DNA Methylation Modulates miRNA Expression in Gonadotroph Pituitary Tumors” or “DNA Methylation Influences miRNA Expression in Gonadotroph Pituitary Tumors”. This is only a suggestion;
2) In line 43 of the Introduction section, substitute “don’t” with the appropriate form “do not”; In line 331 the authors write “lluminaHumanMethylation450k.db” instead of “IlluminaHumanMethylation450k.db”. Please check and correct these and other occurring errors;
3) In the results section, the authors state “Sequencing of small RNA libraries generated an average of 2,497,367 reads per sample, which were mapped to the human genome (hg19) and used for quantification of expression levels of known miRNAs (according to miRBase release 1813).”. It is not clear what version of miRBase was used. Has the latest version been used (Release 22.1)? No “miRBase release 1813” exists. Please clarify;
4) In the subheading “2.3. Validating the role of selected aberrantly methylated miRNAs in a large patients cohort” the authors should clarify why the validation analyses of methylation and miRNAs expression were performed only for the following 4 miRNAs “MIR145, MIR21, MIR184 and MIR134”;
5) I suggest to insert a new Figure 3 reporting the scatter plot graph of miRNAs expression according to methylation levels;
6) One of the major concerns of the present study is the absence of miRNAs evaluation in normal pituitary samples. The authors should have included normal pituitary specimens in order to perform a differential analysis between the expression levels of miRNAs in PitNET samples versus the healthy pituitary tissue counterpart. In this way they would have identified the miRNAs actually involved in tumor development and with altered expression levels. Subsequently, for these miRNAs, the analysis of methylation levels would have further clarified the oncogenic role of these miRNAs and the importance of epigenetic modifications in the mechanisms of neoplastic transformation. Please, clarify why normal pituitary samples were not included and the authors’ opinion about this important issue;
7) In the sub-heading “4.6. miRNA target-predicting”, please clarify why both mirDIP and miRTarbase analyses were performed. Is it necessary to validate the predictions obtained with mirDIP with another software like miRTarbase? Please clarify. In addition, the subheading should be re-named as follows: “4.6. miRNA target-prediction”;
8) In table S5 the authors report all the miRNA-targeted genes. Some genes (e.g. AKIRIN1, BHLHE41, CADM2, SYNM, ZFPM2, etc.) or gene families (e.g. SLC family, SCN family, FAM family, etc.) are targeted by multiple miRNAs. The authors should explore the functional role of these genes modulated mostly by miRNAs;
9) In the Introduction or Discussion sections, the authors should better argue the importance of miRNAs dysregulation in cancer by describing the different studies that identified alterations in the expression levels of the same miRNAs analyzed in this study. For this purpose, see the following and other manuscripts:
- 10.3892/or.2019.7215
- 10.3892/ol.2016.5462
- 10.3892/mmr.2019.9949
- 10.20517/cdr.2019.68
- 10.1159/000113489
Author Response
Dear Editors
Dear Reviewers
We appreciate the valuable comments to our article. The manuscript has been revised according to the suggestions. We made our best to improve the quality of manuscript. The most important change that was introduced is the correction of statistical evaluation due to multiple testing as suggested by Reviewer 2. This was not performed previously for the correlation analysis described in the article. P value adjustment procedure was applied and it notably affected the results and discussion. The number of the CpG sites that are correlated with expression of particular miRNA must to be reduced in the description of results. The interpretation of mRNA target prediction was modified. Details are presented in the reply to comment from Reviewer 2. Please find the replies to all the comments below. A few errors has been corrected in revised manuscript including the description of Table 2.
Reviewer 1
Comments and Suggestions for Authors
Boresowicz and colleagues presented a very interesting research article aimed to determine the role of DNA methylation as an epigenetic mechanism responsible for the modulation of microRNAs expression levels in gonadotroph pituitary neuroendocrine tumors (PiTNETs). Through several experimental and computational approaches, the authors smartly elucidate the negative and positive correlations existing between DNA methylation (both promoter and intragenic methylation) and the expression levels of miRNAs detected in PiTNET samples. Overall, the manuscript is well written and the experimental design well-conceived, however, there are some minor/major revisions that the authors should address:
Comment 1) The title should be modified as follows: “DNA Methylation Modulates miRNA Expression in Gonadotroph Pituitary Tumors” or “DNA Methylation Influences miRNA Expression in Gonadotroph Pituitary Tumors”. This is only a suggestion;
Reply We agree with this suggestion. The title of the revised manuscript is “DNA methylation influences miRNA expression in gonadotroph pituitary tumors”.
Comment 2) In line 43 of the Introduction section, substitute “don’t” with the appropriate form “do not”; In line 331 the authors write “lluminaHumanMethylation450k.db” instead of “IlluminaHumanMethylation450k.db”. Please check and correct these and other occurring errors;
Reply. We made an attempt to correct all the errors.
Comment 3) In the results section, the authors state “Sequencing of small RNA libraries generated an average of 2,497,367 reads per sample, which were mapped to the human genome (hg19) and used for quantification of expression levels of known miRNAs (according to miRBase release 1813).”. It is not clear what version of miRBase was used. Has the latest version been used (Release 22.1)? No “miRBase release 1813” exists. Please clarify;
Reply. This sentence “according to miRBase release 1813” is a mistake. Small RNA sequencing was done in 2018, before release of miRBase v. 22 and reads were annotated using miRBase v. 21. However we believe this has no effect on the results presented in our article since all the miRNA that are covered by HumanMethylation 450K methylation arrays were cataloged in miRBase v. 21
Comment. 4) In the subheading “2.3. Validating the role of selected aberrantly methylated miRNAs in a large patients cohort” the authors should clarify why the validation analyses of methylation and miRNAs expression were performed only for the following 4 miRNAs “MIR145, MIR21, MIR184 and MIR134”;
Reply. We were not able to investigate all the miRNAs of interest. We selected a few. In the revised manuscript these are MIR145, MIR21 and MIR184. (MIR134 was removed because CpG cg13753460 did not met criterion of the adjusted p-value when correction for multiple comparisons was applied according to suggestion from Reviewer 2). MIR145 and MIR21 were chosen since both has a documented role in pituitary tumors. MIR184 was the found as overexpressed in pituitary tumors in previous reports and this is in line with fact that we observe DNA hypomethylation at MIR184 and negative methylation/expression correlation for this miRNA. This justification was introduced in the revised manuscript.
Comment 5) I suggest to insert a new Figure 3 reporting the scatter plot graph of miRNAs expression according to methylation levels;
Reply. We introduced Figure 2B that shows matched result of DNA methylation levels at particular CpG sites and miRNA expression levels.
Comment 6) One of the major concerns of the present study is the absence of miRNAs evaluation in normal pituitary samples. The authors should have included normal pituitary specimens in order to perform a differential analysis between the expression levels of miRNAs in PitNET samples versus the healthy pituitary tissue counterpart. In this way they would have identified the miRNAs actually involved in tumor development and with altered expression levels. Subsequently, for these miRNAs, the analysis of methylation levels would have further clarified the oncogenic role of these miRNAs and the importance of epigenetic modifications in the mechanisms of neoplastic transformation. Please, clarify why normal pituitary samples were not included and the authors’ opinion about this important issue;
Reply. We agree that including the sequencing analysis of miRNA in normal pituitary gland would provide some additional important information in this article. This analysis were originally planned for the study.
The main problem is the availability of normal human pituitary samples that could be used for the molecular analysis. Normal pituitary samples can not be obtained from patients or any healthy donors in compliance with ethical standards. We have collected normal pituitary gland sections from autopsies. In contrary to tumor samples that are snap frozen immediately after tissue removal the normal samples from autopsy are collected hours after the death of the particular persons. This post mortem interval (PMI) for the samples used in the study is between 21 to 46 hours. According to a published data such PMI doesn’t have a notable impact on DNA methylation analysis [1] [2]. Some degradation related artefacts in DNA methylation measurements may appear in samples collected at 72 hour after death (or later) where DNA degradation was also observed on the gel [1].
In contrary to DNA methylation analysis transcriptomic profiling can be strongly affected by delay in sample collecting after death. Degradation of RNA which progress within a minutes after death [3] has probably a limited effect on miRNA profile which are belived as relatively stable molecules. The changes of transcription after death is not only a result of RNA degradation but also it represent response to a change of philological conditions [4]. It was recently shown that prolonged delay in fixation affect results of small RNA sequencing [5]. The level of particular RNAs including miRNAs [6] can be even used as a marker for testing post mortem interval [7].
For these reasons we decided to use the post mortem samples for DNA Methylation analysis, not for transcriptomic profiling. This explanation is provided in the manuscript: “Five samples of normal human pituitary from autopsies were used for DNA methylation profiling. The samples underwent hematoxylin/eosin staining and histopathological evaluation to exclude the presence of incidental pituitary tumors. Postmortem interval (PMI) for the samples was in the range of 25-46 hours (median 35,5). According to previously reported data this PMI doesn’t affect DNA methylation assessment with bisulfite-based methods [45], but samples with this PMI are not suitable for reliable expression analysis profile [46] thus normal pituitary samples were used for DNA methylation only.” The reference 46 has been updated in revised manuscript.
We made an attempt to overcome problem of using postmortem normal tissue by the use of archival formalin-fixed pituitary sections. According to the materials provided by the sequencing technology supplier the miRNA sequencing results on FFPE samples are highly correlated with those from fresh frozen. We used archival samples of normal pituitary obtained from surgery of Rathke's cleft cyst which is s a rare benign growth on the pituitary gland. This samples represent histologically normal pituitary and could be used as a control and were subjected to isolation of total RNA and sequencing of small RNA fraction as tumor samples. Unfortunately the analysis of sequencing data showed that including FFPE derived RNA samples strongly biased the normalization of data and output of the analysis. Unfortunately, we couldn’t combine FFPE and fresh frozen tissue in the analysis to get any reliable results. It was highly probable that the difference that we observed between normal and tumors are due to a tissue preservative method not physiological conditions.
Comment 7) In the sub-heading “4.6. miRNA target-predicting”, please clarify why both mirDIP and miRTarbase analyses were performed. Is it necessary to validate the predictions obtained with mirDIP with another software like miRTarbase? Please clarify. In addition, the subheading should be re-named as follows: “4.6. miRNA target-prediction”;
Reply. The difference between MirDIP and miRTarbase is that the first one combines the tolls for predicting miRNA-mRNA interactions based on predicting algorithms and the second tool is a database of the experimentally observed interactions. We believe that miRTarbase can provide some complementary information to basic target prediction with MirDIP. This explanation was introduced in main text. The subheading was re-named according to the suggestion
Comment 8) In table S5 the authors report all the miRNA-targeted genes. Some genes (e.g. AKIRIN1, BHLHE41, CADM2, SYNM, ZFPM2, etc.) or gene families (e.g. SLC family, SCN family, FAM family, etc.) are targeted by multiple miRNAs. The authors should explore the functional role of these genes modulated mostly by miRNAs;
Reply According to the suggestion from Reviewer 2 we modified the statistical approach, and included the p-value adjustment due to multiple testing in the correlation analysis. This resulted in a decrease of the number of miRNA that can be interpreted as DNA methylation related since part of CpG sites were below statistical level. We also used more rigorous criterion in correlation analysis in mRNA target prediction and we obtained a reduced number of potential targets. Parts of potential mRNA targets were removed from the results section and are not commented in the discussion.
Comment 9) In the Introduction or Discussion sections, the authors should better argue the importance of miRNAs dysregulation in cancer by describing the different studies that identified alterations in the expression levels of the same miRNAs analyzed in this study. For this purpose, see the following and other manuscripts:
- 10.3892/or.2019.7215
- 10.3892/ol.2016.5462
- 10.3892/mmr.2019.9949
- 10.20517/cdr.2019.68
- 10.1159/000113489
Reply. This issue was more comprehensively commented in discussion section of revised manuscript. Since there is a very large number of the publications on the cancer-related role of our miRNAs of interest we referenced mainly to most recently published reviews and selected articles. This part of discussion is difficult because a large number of reports has been published and inconsistent results were found in studies on different tumors. Frequently the same miRNA act as oncogene in one cancer type and tumor suppressor in other one, and sometimes the contradictive functions are reported for the same cancer type. We would rather to mark this issue in the discussion section and not to explore it deeply.

Reviewer 2 Report
This is a very interesting study that aimed to understand the contribution of DNA methylation to miRNA expression in gonadotroph pituitary tumors. Overall, the manuscript is very well presented, the methodology is appropriate and the conclusions are supported by the data. The only major concern is that statistical tests should have been corrected for multiple testing to avoid false positives. Adjusted p value is mentioned only for pathway enrichment in gene ontology catalogs and in the definition of differentially methylated probes. The latter considers adjusted p>0.05, which is not intuitive. This should be clarified. The type of correcting method for multiple testing, such as Benjamini & Hochberg or Bonferroni, should be indicated and applied throughout the manuscript.
Author Response
Dear Editors
Dear Reviewers
We appreciate the valuable comments to our article. The manuscript has been revised according to the suggestions. We made our best to improve the quality of manuscript. The most important change that was introduced is the correction of statistical evaluation due to multiple testing as suggested by Reviewer 2. This was not performed previously for the correlation analysis described in the article. P value adjustment procedure was applied and it notably affected the results and discussion. The number of the CpG sites that are correlated with expression of particular miRNA must to be reduced in the description of results. The interpretation of mRNA target prediction was modified. Details are presented in the reply to comment from Reviewer 2. Please find the replies to all the comments below. A few errors has been corrected in revised manuscript including the description of Table 2.
Reviewer 2
Comments and Suggestions for Authors
This is a very interesting study that aimed to understand the contribution of DNA methylation to miRNA expression in gonadotroph pituitary tumors. Overall, the manuscript is very well presented, the methodology is appropriate and the conclusions are supported by the data.
Comment The only major concern is that statistical tests should have been corrected for multiple testing to avoid false positives. Adjusted p value is mentioned only for pathway enrichment in gene ontology catalogs and in the definition of differentially methylated probes.
The latter considers adjusted p>0.05, which is not intuitive. This should be clarified. The type of correcting method for multiple testing, such as Benjamini & Hochberg or Bonferroni, should be indicated and applied throughout the manuscript.
Reply. In the analysis of correlation p-value correction was not applied. This has been done in the revised manuscript and notably affected the interpretation of our results.
As the result of p-value adjustment we had to exclude part of CpG correlated with miRNA expression level, since adjusted results were below significance threshold
-The results section 2.2 was reedited and 12 CpGs described previously as correlated with miRNA expression were excluded from the results in the revised manuscript.
- Table S3 was modified
- Keeping the criterion of adjusted p-value we had to remove hsa-miR-134-5p from validation analysis since the CpGs lecated within this miRNA didn’t met adj p<0.05 in analysis of DNA methylation and correlation. The changes in Results section 2.3 were introduced.
- Target prediction procedure was limited only to those miRNA that met adj p<0.05 in analysis of DNA methylation and correlation. Therefore results section 2.4 was reedited.
- Discussion section was modified according to the changes in the description of results
We used adjustment for multiple testing in the correlation analysis included in target reduction procedure. To reduce the number of multiple test we modified criterion of miRNA-mRNA prediction score. In the revised manuscript only miRNA-mRNA pairs with VeryHigh probability of interactions were taken into account. In spite of this modification any of correlation results remained significant after p-value adjustment. However we believe that the results that are presented in revised manuscript may still provide the important information: they include only highly probable target prediction, part of the predicted mRNA targets are experimentally validated according to miRTarbase database, the correlation coefficient is below -0.5 for all the miRNA-mRNA target predicted pairs. Since the correlation analysis is only a part of predicting the possible miRNA-mRNA interactions we decided to leave the description of the results in the revised manuscript.
In the revised text we present the results of the analysis with a clear highlighting that correlation analysis were not significant when correction for multiple testing was applied and the p-value presented in Table 2 is not adjusted. Table S4 was modified.
P-value adjustment is clearly indicated in Material and Methods/4.7 Statistical analysis in the revised manuscript.
References
- Rhein, M.; Hagemeier, L.; Klintschar, M.; Muschler, M.; Bleich, S.; Frieling, H. DNA methylation results depend on DNA integrity- role of post mortem interval. Front. Genet. 2015, 6, 1–7.
- Barrachina, M.; Ferrer, I. DNA Methylation of Alzheimer Disease and Tauopathy-Related Genes in Postmortem Brain. J. Neuropathol. Exp. Neurol. 2009, 68, 880–891.
- Sidova, M.; Tomankova, S.; Abaffy, P.; Kubista, M.; Sindelka, R. Effects of post-mortem and physical degradation on RNA integrity and quality. Biomol. Detect. Quantif. 2015, 5, 3–9.
- Ferreira, P.G.; Muñoz-Aguirre, M.; Reverter, F.; Sá Godinho, C.P.; Sousa, A.; Amadoz, A.; Sodaei, R.; Hidalgo, M.R.; Pervouchine, D.; Carbonell-Caballero, J.; et al. The effects of death and post-mortem cold ischemia on human tissue transcriptomes. Nat. Commun. 2018, 9.
- Helen, M.M. Deleterious effects of formalin- fixation and delays to fixation on RNA and miRNA-Seq profiles. 2019, 1–10.
- Tu, C.; Du, T.; Ye, X.; Shao, C.; Xie, J.; Shen, Y. Using miRNAs and circRNAs to estimate PMI in advanced stage. Leg. Med. 2019.
- Sharma, S.; Singh, D.; Kaul, D. AATF RNome has the potential to define post mortem interval. Forensic Sci. Int. 2015, 247, e21–e24.
- Pozhitkov, A.E.; Neme, R.; Domazet-Lošo, T.; Leroux, B.G.; Soni, S.; Tautz, D.; Noble, P.A. Tracing the dynamics of gene transcripts after organismal death. Open Biol. 2017, 7, 160267.
